# Modeling sickness absence data: A scoping review

**Tom Duchemin**[1,2]*, **Mounia N. Hocine**[1]

**1** Laboratoire Modélisation, Epidémiologie et Surveillance des Risques Sanitaires, Conservatoire national des arts et métiers, Paris, France, **2** Malakoff Médéric Humanis, Paris, France

* tom.duchemin@cnam.fr

## Abstract

The identification of sick leave determinants could positively influence decision making to improve worker quality of life and to reduce consequently costs for society. Sick leave is a research topic of interest in economics, psychology, health and social behaviour. The question of choosing an appropriate statistical tool to analyse sick leave data can be challenging. In fact, sick leave data have a complex structure, characterized by two dimensions: frequency and duration, and involve numerous features related to individual and environmental factors. We conducted a scoping review to characterize statistical approaches to analyse individual sick leave data in order to synthesise key insights from the extensive literature, as well as to identify gaps in research. We followed the PRISMA methodology for scoping reviews and searched Medline, World of Science, Science Direct, Psycinfo and EconLit for publications using statistical modeling for explaining or predicting sick leave at the individual level. We selected 469 articles from the 5983 retrieved, dated from 1981 to 2019. In total, three types of model were identified: univariate outcome modeling using for the most part count models (438 articles), bivariate outcome modeling (14 articles), such as multistate models and structural equation modeling (22 articles). The review shows that there was a lack of evaluation of the models as predictive accuracy was only evaluated in 18 articles and the explanatory accuracy in 43 articles. Further research based on joint models could bring more insights on sick leave spells, considering both their frequency and duration.

## 1. Introduction

Understanding sick leave (SL) is a crucial issue for workers and their employers. Identification of determinants of sick leave could help decision makers set up appropriate prevention policies to improve the quality of life of workers and reduce costs for employers [1, 2]. However, potential determinants are diverse, and their identification may be extremely difficult. SL may be related to both individual and professional environments [3, 4], and indeed, the literature covering the topic is very wide-ranging. Studies can be found in journals of public health, epidemiology, sociology, economics, and psychology [5–8]. In addition, the modeling of SL is made difficult by certain features of the collected data. For instance, SL data can be zero-inflated, over-dispersed, censored, truncated, or highly seasonal, to give a few examples. Furthermore,

**Data Availability Statement:** Supplementary references (indexed by A in the text) and data from the scoping review are available at 10.6084/m9.figshare.9741203.

**Funding:** The Ph.D. works of TD are funded by a 3-year grant from the Association Nationale de la

Recherche et de la Technologie (grant 2017/1517) and by Malakoff médéric humanis. The funders had no role in study design, data collection and analysis, decision to publish, or preparation of the manuscript.

**Competing interests:** The authors have declared that no competing interests exist.

SL is a two-dimensional variable characterised by both its frequency (number of SL spells over a given period) and its duration (length of a SL spell). Thus, the choice of appropriate statistical tools is very important.

Here, we document the state of the art on statistical methods for modeling individual SL data. This may help identify major trends and gaps in the scientific literature that could guide researchers towards better modeling. We pay careful attention to the afore-mentioned issues, summarize the results from the literature, and describe the best-adapted statistical approaches to deal with the properties of SL data. To proceed, we followed the *scoping review* methodology recently published by PRISMA [9]. While reviews on the determinants of SL have been previously published [4, 10–13], this is, to our knowledge, the first review providing an overview of the various statistical tools that can be used to identify SL determinants.

## 2. Methods

### 2.1. Literature search

We performed a scoping literature review to assess how SL is modeled. A scoping approach was preferred to a systematic approach in order to describe trends and practices in different fields of research, and formulate new research possibilities. The recently published *PRISMA* (Preferred Reporting Items for Systematic reviews and Meta-Analysis) extension for scoping reviews *(PRISMA-ScR)* [9] was used.

To perform a comprehensive search, we systematically searched five databases from different fields which may deal in SL analyses: *MEDLINE*, *World of Science*, *Science Direct*, *PsycInfo* and *EconLit*. We formulated queries on three items: *(i)* model (keywords like *explaining*, *predicting*, *factor*, *determinant*, *risk*, *model*, and *classification*), *(ii)* absence (keywords like *sickness absence*, *absenteeism*, *sickness spell*, *sick leave* and *sick-leave*) and *(iii)* work (keywords like *working*, *worker*, *employee* and *adult*). The *MEDLINE* query was:

*(predic$[Title/Abstract] OR risk$[Title/Abstract] OR classification$[Title/Abstract] OR regression$[Title/Abstract] OR explain$[Title/Abstract] OR determinant$[Title/Abstract] OR factor$[Title/Abstract] OR model$[Title/Abstract])*
*AND*
*(work$[Title/Abstract] OR employee$[Title/Abstract] OR adult$[Title/Abstract])*
*AND*
*(sickness absen$[Title/Abstract] OR sickness spell$[Title/Abstract] OR sick leave$[Title/Abstract] OR sick-leave$[Title/Abstract] OR absenteeism[Title/Abstract])*

Searches were performed in September 2019 over the full databases.

The review was then performed in two steps. The first consisted in a review of titles and abstracts by a single reviewer (T.D.). Articles were included if they met the following criteria:

1. Original articles published in peer-reviewed journals (which excludes theses, book chapters, conference communications, reviews, etc.).

2. Involve statistical models describing any outcome related to the occurrence of SL at the individual level. Hence, any models describing aggregated data were not considered relevant.

3. Explicitly mention sickness absence in a working population as an outcome of a statistical model.

We chose to exclude articles based on aggregated data for two reasons. The first reason is that aggregated data does not have the same meaning as individual data: models for aggregated data gives information about populations' behavior and not about individuals' and we decided

to only talk about individuals. The second reason is that those data have a different structure than individual data on SL and may require different modeling.

When a reference could not be included with certainty, its full text was obtained and a second screening was performed. This second step consisted of a review of the articles' contents by two reviewers (T.D. and M.H.).

The first step was performed by a single reviewer because the amount of retrieved material was overwhelming: this first step is therefore not infallible, and mistakes may have been made. However, we have tried to reduce these errors by eliminating only articles that seemed clearly inappropriate during this first step, and by keeping all items that may have caused any doubt. We considered as inappropriate articles that did clearly not deal with sick leave or that provided only qualitative analyses.

For the sake of clarity, articles retrieved from the review are referenced with the prefix A in the *Results* section and can be found in the Electronic Supplementary Material. This document also provides all the results from this scoping review.

## 2.2. Resource extraction and analysis

The informations extracted from each included article consisted of:

1. *Metadata*: publication year, journal title, authors;

2. *Dataset characteristics*: study population, country of the study, first year of the study;

3. *Statistical methods*: statistical model, definition of the outcome, criteria for the evaluation of the models, predictive ability of the model (AUC or C-Index), consideration of the multi-level nature of SL data.

We only retained the value of the evaluation criterion for predictive models because the AUC and C-Index are comparable; even if the statistical methodologies are different. Note that explicative criteria such as $R^2$ are dependent on the model and are thus not relevant for model comparison in our scoping review, where we describe different methods. Finally, descriptive analyses were performed on the extracted data to examine frequencies and trends of statistical approaches in the literature.

## 3. Results

### 3.1. Study selection

A total of 5150 articles were retrieved from *MEDLINE*, 226 from *Science Direct*, 419 from *World of Science*, 173 from *PsycInfo*, and 15 from *EconLit*. After screening abstracts and titles, we excluded 3536 publications that did not meet our inclusion criteria. By examining the remaining 885 in detail, a further 416 were excluded. Thus, a total of 469 articles were retained for the scoping literature review. Fig 1 shows the PRISMA flow diagram for the study's selection method.

The first article modeling SL identified in our selection was published in 1981 [A358]. While SL articles go back to at least 1958 [14], the earliest ones only used descriptive tools, not statistical models. Fig 2 shows the number of published peer review articles retained per year: very few articles modeling SL with statistical tools were published before 1998. Since then, the number has tended to increase over time.

Retained articles were mainly published in occupational health journals, as shown in Table 1. However, as SL is an interdisciplinary research theme, journals in the fields of public health, economics, social science, general medicine, psychology, and generalist journals, are also represented. We note that almost half of the databases used for analyses comes from

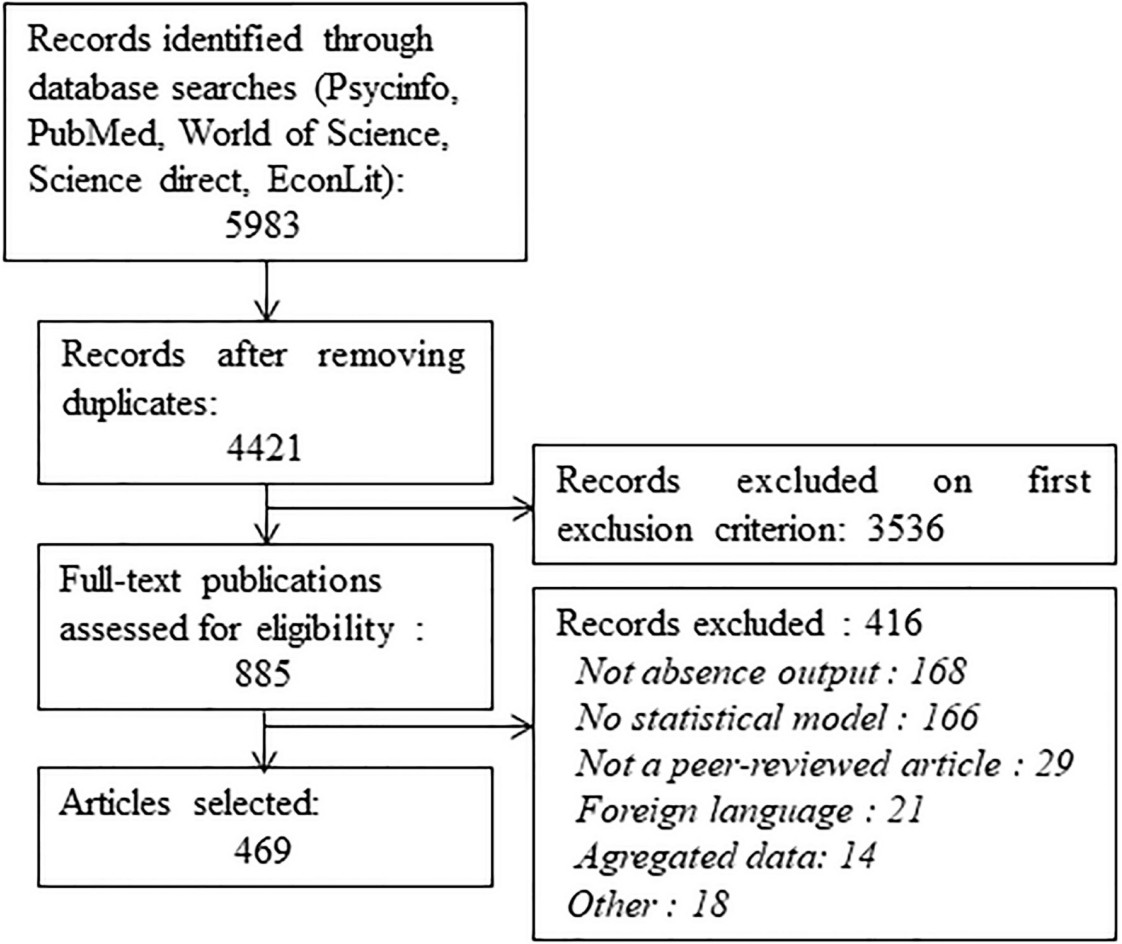

**Fig 1. PRISMA diagram of the selection process for study inclusion in the review.**

Scandinavian countries, with 231 articles publishing studies on databases from Finland, Denmark, Sweden, and Norway. Databases from the Netherlands and the US are analyzed in 64 and 28 publications, respectively. Databases from remaining countries correspond to less than 20 articles per country.

Retrieved publications used two different kinds of data source for their analyses: *(i)* administrative databases or registers describing certified SL of workers, and *(ii)* questionnaires describing declared SL. Across the 469 studies, the formers are found in 346 studies, the latter in 421. In 302 articles, register and questionnaire were linked together. One article also used data from a meta-analysis.

A total of 425 publications focused on all-cause SL, while the remaining few focused on precise causes of SL. Among those, 11 with low back pain-related SL, 12 dealt with musculoskeletal disorder-related SL, 12 with depressive or mental disorder-related SL, and 10 with other cause-specific SL such as voice problems, work-related SL, or respiratory complaints. The retrieved articles many different populations with specific job, specific pathologies or more generally specific characteristics. The most studied population is a general population without specific characteristics (231 articles). A very large number of articles also study healthcare

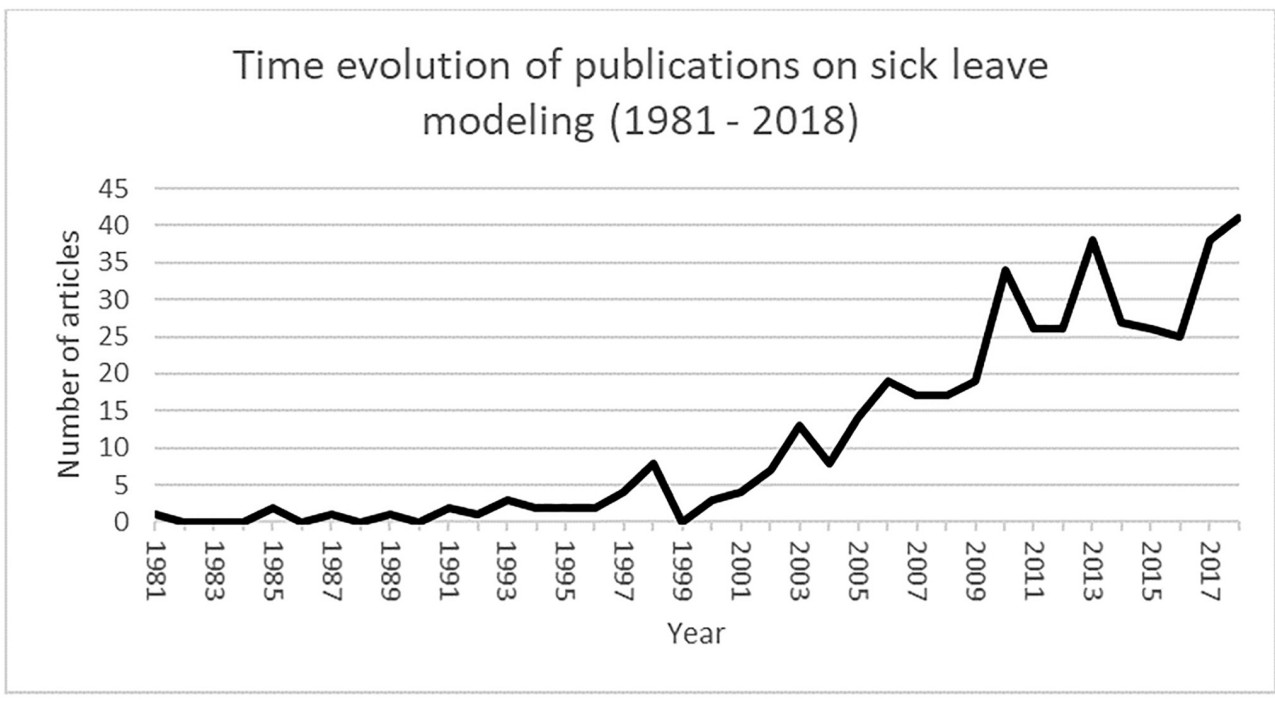

**Fig 2. Number of articles included in the scoping literature review from each year.**

workers (75 articles) and employees in the public sector, healthcare workers excluded (43 articles).

## 3.2 Statistical methods for modeling sick leave

Three main categories of statistical approach for modeling SL arise from the results of the review, as illustrated in Table 2:

1. *Modeling a univariate SL outcome*: based on regression models, this allows researchers to evaluate the effect of potential predictors. In this approach, SL data of various forms are

**Table 1. Number of retrieved articles per peer reviewed journal.**

| Journal | Number (%) |
|---|---|
| *Journal of Occupational and Environmental Medicine* | 38 (9,5%) |
| *Occupational and Environmental Medicine* | 33 (8,25%) |
| *International Archives of Occupational and Environmental Health* | 26 (6,5%) |
| *Scandinavian Journal of Work, Environment & Health* | 26 (6,5%) |
| *BMC public health* | 24 (6%) |
| *Scandinavian Journal of Public Health* | 21 (5,25%) |
| *European Journal of Public Health* | 19 (4,75%) |
| *Occupational Medicine (Oxford, England)* | 19 (4,75%) |
| *Social Science & Medicine* | 11 (2,75%) |
| *Journal of Occupational Rehabilitation* | 12 (3%) |

**Table 2. Methods retrieved from the scoping review.**

| Outcome of interest | | | Model |
|---|---|---|---|
| **Modeling a univariate SL outcome (n = 438)** | *Count data (n = 132)* | | • Poisson (84)<br>• Negative binomial (44)<br>• Zero-inflated negative binomial (9)<br>• Hurdle model (8)<br>• Zero-inflated Poisson (2) |
| | *Time-to-event (n = 86)* | | • Cox (77)<br>• Andersen-Gill (2)<br>• Frailty model (1)<br>• Competing risk model (1)<br>• Other parametric (3) or nonparametric (2) |
| | *Categorical data (n = 190)* | *Binary (n = 171)* | • Logit (167)<br>• Probit (3)<br>• GLM with unspecified link (2) |
| | | *Multinomial (n = 20)* | • Multinomial logistic (12)<br>• Ordinal logistic (8) |
| | *Rate/Continuous indicator (n = 57)* | | • Linear regression (45)<br>• Joinpoint (2)<br>• Tobit (1)<br>• Fuzzy network and machine learning methods (1)<br>• GLM with unspecified link (2) |
| **Modeling a bivariate SL outcome (n = 14)** | | | • Multistate model (7)<br>• Trajectory analysis (7) |
| **Structural equation modeling (n = 22)** | | | |

considered, involving a number of statistical approaches, as described below. A total of 372 publications included at least one of those methods:

a. *Count data*: SL data can be defined as the number of days (or hours) of absence, or the number of SL spells during a given time interval. To model count data, the authors use Poisson models (84 articles,) [A1-A84], negative binomial models (50 articles) [A1-A7, A85-A121], zero-inflated negative binomial models (9 articles) [A120-A128], hurdle models (10 articles) [A117-A120,A129-A132], and zero-inflated Poisson models (2 articles) [A128,A133]. These methods are described in 141 articles of those retained in the present review.

b. *Time-to-event data*: SL data can be defined in terms of time to subsequent SL. This approach is used in 86 publications. Of these, 77 use Cox models [A17,A26,A120-A208]. Four publications use models derived from Cox models: the Andersen-Gill model [A209,A210], frailty modeling [A67], and competing risk models [A211]. Other publications use parametric models [A46,A212,A213] or other nonparametric models [A214, A215].

c. *Categorical data*: SL data can be defined in terms of experiencing at least one SL spell, or another predefined categorical SL event, over a given time interval. SL is described using a binary outcome in 185 publications, and with a multiple category outcome in 24 publications. For binary SL outcomes, 181 models use a logit link function [A9,A80,A90,A96, A102,A149,A216-A376], 3 a probit link function [A6,A376,A377], and 2 use GLM with

an unspecified canonical link [A378,A379]. For multinomial SL outcomes, 15 publications use multinomial logistic regression [A188,A232,A380-A389] and 8 use ordinal logistic regression [A391-A397].

 d. *Continuous indicators*: SL data can also be defined as the number of days, spells, or hours, the sickness absence rate, the average duration of spells, etc. A total of 57 publications involve a continuous SL indicator. Amon them, 51 use linear regression [A4,A20, A65,A90,A98,A108,A177,A331,A398-A434], 2 use joinpoint regression [A435,A436], 1 uses Tobit regression [A437], and 2 use GLM with an unspecified link function [A4, A438]. Another article uses a deep learning method, a fuzzy network model, and compares it to machine learning method [A439].

2. *Modeling a bivariate SL outcome*: based on joint models, this strategy allows researchers to evaluate the effect of potential predictors on both SL duration and frequency. Seventeen publications use a joint approach, modeling simultaneously the duration and frequency of SL. Eight of these use multi-state modeling [A189,A440-A445], while the nine others use trajectory analysis [A13,A42,A385,A446-A449].

3. *Structural equation modeling (SEM)*: based on a factorial analysis and a linear regression model, this allows testing, without quantifying, the causal association between predefined latent variables and SL. In this approach, SL is defined as a continuous outcome: frequency, duration, or a combination of both. A total of 23 publications use this approach [A277, A416,A450-A469], mostly in journals involving qualitative research such as *the Journal of Organizational Behavior* [A458-A461].

We finally found that 52 articles evaluate the accuracy of these models in terms of predictability or explainability. Among those 5752 18 evaluated the predictive capacity of their models using AUC or the C-Index. Obtained AUCs ranged from 0.53 to 0.88. Further, 43 publications among those 57 proposed an evaluation of the models using $R^2$ or pseudo $R^2$ for regression models, or with specific criteria for conceptual models (e.g., AGFI, RMSEA, IFI, NNFI, CFI). Moreover, 51 of the 469 articles took into account the multilevel nature of SL data by using hierarchical modeling or by including mixed effects to describe both individual and work levels.

## 4. Discussion

This scoping review has highlighted the increase in statistical modeling of sick leave data over the previous 40 years. We identified different statistical approaches which could be gathered into three categories: models for univariate SL outcome, models for bivariate SL outcome, and structural equation modeling. Logistic regression and count data models are the most popular approaches: logistic regression gives intuitive results thank to odd ratios when the problem involves a binary outcome; count data models are also often used to analyse variables such as number of day of SL or number of spells per year. Most of the time, the publications focus on non-specific population, but many articles focus on the case of healthcare workers and civil servants, who are very affected by sickness absenteeism. The way outcomes are encoded can vary greatly (occurrence of a spell, frequency, duration, etc.). Finally, the predictive performance of SL models also varies a lot from one publication to the next: when provided, the AUC and C-Index values range from 0.53 to 0.88.

This review was faced with the difficulty of dealing with the huge number of publications that exist on SL. We have tried to be as exhaustive as possible by processing data from very

numerous sources and we retrieved a huge amount of publications. Consequently, the first step of the review was carried out by a single reviewer and we may have failed to identify all relevant articles during this first step of the review. Finally, we chose to perform a scoping review but a systematic review of the literature could have been interesting and would have more comprehensibly assessed the quality of suggested models. However, the simultaneous evaluation of so many different models would have been overly complex, and a scoping review approach seems more appropriate.

We would particularly like to point out the lack of information provided to assess the predictability and explainability of models on SL data. First, very few articles evaluate the accuracy of their models in terms of explainability and predictability. This is however crucial, as SL is a phenomenon with many potential determinants: accuracy criteria could help researchers focus on determinants that are more relevant than other. Second, many of the accuracy scores obtained are fairly poor, as mentioned earlier. A possible reason could be the fact that SL is likely related to many determinants, but most studies investigate them partially. A systematic review of the determinants of SL would be very valuable: the SL literature is abundant but nevertheless, the mechanisms of sick leave remain unclear. This said, such an exercise would be complex since, as explained above, definition of outcomes can be very different. Indeed, distinct reviews would be needed for frequency and duration analyses.

Data collection is also another point to improve the predictability of sick leave. Indeed, every company produces sick leave data and it could be a powerful way to deal with sick leave analysis. For instance, an article retrieved in this review uses neural networks to predict sick leave for this type of administrative data and its results seem encouraging [A439] [15]. These models work well for massive data set, both in terms of number of observations and of covariates and a more systematic collection and analysis of those data could lead to a better understanding of the phenomenon.

Another way to improve the explainability of SL would be to investigate alternative statistical approaches. In addition to a systematic review on determinants, new methods for hierarchizing SL determinants could be helpful in providing new insights on them. By introducing a great number of heterogeneous variables into a model, methods such as random forests might more easily identify the most important variables [16]. Such methods may be appropriate for the study of sick leave because the possible determinants of SL are numerous and these methods, running without linearity hypotheses, may potentially better explain SL. Those models could also take easily into account the multilevel nature of SL data by implicitly testing all interactions. The high correlation between the predictors of sick leave are moreover a key issue that is seldom discussed in the literature. Most of the times, few covariates are used and the correlations do not really impact the methodology in this case: it will mostly impact the interpretation of the results. However, when using lots of covariates, the high correlation becomes an issue. We had noted an article using variable selection [17]. This article used a backward selection to choose the most relevant variable. This allows for a more readable model but do not really solve the problem of correlations. If variables are highly correlated, the model will simply choose the most significant variables from this bunch of correlated variables. Variable selection is necessary if there are too many covariates but must always be coupled with a good correlation analysis.

Moreover, only 19 articles investigated causal relationships between covariates and SL using structural equation modeling, despite this approach seeming rather informative for the study of SL. Indeed, since the most commonly used regression methods described in this scoping review give fairly poor explanatory results, structural equation modeling could be used to effectively and intuitively test the links between different variables. Bayesian network models, none of which were identified in this review, might be an appropriate and powerful solution

for causal inference [18, 19]. Furthermore, models for bivariate outcomes, and in particular multi-state models, also appear to have been overlooked in previous research. Indeed, most of the retrieved articles deal with SL in a single dimension (duration or frequency), yet both are of interest and potentially correlated. A joint study of the two could make it possible to improve the explanatory accuracy of models, while also helping to better understand the causes and features of SL.

## Supporting information

**S1 Text.**
(TXT)

**S1 Checklist. PRISMA-ScR checklist.**
(DOCX)

## Author Contributions

**Conceptualization:** Tom Duchemin, Mounia N. Hocine.

**Data curation:** Tom Duchemin.

**Formal analysis:** Tom Duchemin.

**Investigation:** Tom Duchemin, Mounia N. Hocine.

**Methodology:** Tom Duchemin, Mounia N. Hocine.

**Supervision:** Tom Duchemin.

**Validation:** Tom Duchemin, Mounia N. Hocine.

**Writing – original draft:** Tom Duchemin, Mounia N. Hocine.

**Writing – review & editing:** Tom Duchemin, Mounia N. Hocine.

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
