## [Decision Letter · Decision Letter 0]

3 Jan 2020

PONE-D-19-24227

Modeling sickness absence data: a scoping review

PLOS ONE

Dear Mr. Duchemin,

Thank you for submitting your manuscript to PLOS ONE. After careful consideration, we feel that it has merit but does not fully meet PLOS ONE’s publication criteria as it currently stands. Therefore, we invite you to submit a revised version of the manuscript that addresses the points raised during the review process.

We would appreciate receiving your revised manuscript by Feb 17 2020 11:59PM. To enhance the reproducibility of your results, we recommend that if applicable you deposit your laboratory protocols in protocols.io, where a protocol can be assigned its own identifier (DOI) such that it can be cited independently in the future. For instructions see: http://journals.plos.org/plosone/s/submission-guidelines#loc-laboratory-protocols

We look forward to receiving your revised manuscript.

Kind regards,

Beverley J Shea, Ph.D

Academic Editor

PLOS ONE

Journal Requirements:

3. Please amend the file type of your Prisma checklist file from 'other' to 'supporting information'

Reviewers' comments:

Reviewer's Responses to Questions

**Comments to the Author**

1. Is the manuscript technically sound, and do the data support the conclusions?

Reviewer #1: Partly

2. Has the statistical analysis been performed appropriately and rigorously? 

Reviewer #1: No

3. Have the authors made all data underlying the findings in their manuscript fully available?

Reviewer #1: No

4. Is the manuscript presented in an intelligible fashion and written in standard English?

Reviewer #1: Yes

5. Review Comments to the Author

Reviewer #1: Please see my report in the attachment.

Please see my report in the attachment.

Please see my report in the attachment.

Please see my report in the attachment.

Please see my report in the attachment. Please see my report in the attachment.

6. PLOS authors have the option to publish the peer review history of their article (what does this mean?). If published, this will include your full peer review and any attached files.

Reviewer #1: No

---

## [Author Response · Author response to Decision Letter 0]

4 Feb 2020

First of all, we would like to thank the reviewer for his/her comments and questions who has helped to improve the quality of the manuscript thanks to relevant remarks. We have tried to answer them point by point and we hope that our answers are clear and comprehensive.

Major comments

• On page 5, line 81 to 82, in the criterion description, only the individual level involved SL studies were to be included. And models on the aggregated data would be considered irrelevant. Could it be explained why aggregated data on SL study is irrelevant in the search. Aggregated data could be fitted by a model talking about behaviors among different population’s sick leave.

The reviewer’s comment is very relevant: aggregated data are indeed appropriate to study sick leave.

We excluded articles based on aggregated data for two reasons. The first reason is that aggregated data does not have the same meaning as individual data: models for aggregated data gives information about populations’ behavior and not about individuals’ and we decided to only talk about individuals. For instance, we excluded articles evaluating interventions1, predicting the trend of absenteeism at the organizational level2 or evaluating the impact of biometeorological effects on worker absenteeism3. While the latter describes a determinant of absenteeism, the other two are more unclassifiable with respect to our objective. We have therefore decided to exclude all of these articles. The second reason is that those data have a different structure than individual data on SL and may require different modeling such as time series models. It would be difficult to add those models among the three categories we built. We could have built a fourth category, but it seems to us that this category would be somewhat disconnected from the rest of the article and would deserve its own analysis. 

We thank the reviewer for pointing out this issue because it was missing in the article. We have then:

- rewritten our objective (line 47) to specify that we are only studying models for individual sick leave data. 

- explained our choice to exclude those data in the exclusion criteria (lines 86 – 90)

• In Figure 1, what is the relationship between those numbers, some numbers seemed to be able to be added up to get another number but it wasn’t the fact, such as 469+416 is not 877. Could there be more introduction on those numbers?

We would like to thank the reviewer for pointing out these discrepancies. Those numbers have to add up and the numbers presented are incorrect. This is the result of confusion when we add to update the review.

We checked our calculations from the result files and the correct numbers are as follow:

- We retrieved 5983 articles from our queries and 4421 after removing duplicates.

- We excluded 3536 articles at the first step and selected 885 for the second step.

- We included 469 articles in the final analysis and we excluded 416 articles.

The figures and texts have been modified accordingly.

Minor comments

• If an article appears on a peer-reviewed journal, to what extent, this article will be considered to be inappropriate. (Page 5,line 90)

The explanation of this step could indeed be clearer, and we thank the reviewer for this question. We rejected articles that did clearly not deal with sick leave or that provided only qualitative analyses. As the criteria chosen for the request were not very restrictive, we had a lot of articles that mentioned sick leave in the abstract without treating them in a model. 

We added this explanation to our manuscript to clarify our approach (lines 98-99).

• As mentioned, the Sick Leave is an interdisciplinary research theme, so is there any chance that the same model with the same approach will be published in different fields thus causing some duplication of collecting the articles.

We thank the reviewer for this comment. In our opinion, the presence of duplication is not really an issue for the objective of our article. It is indeed a scoping review and we propose bibliometric measures and general conclusions on methodology published by the literature. Even if the article used is the same, the presentation of results and methodological discussions may be different. If our article was a systematic literature review or meta-analysis to assess the impact of this or that factor, this would indeed be a problem. 

Nevertheless, we tried to assess this point to see if duplicate publication was common in the sick leave literature. By studying our database with the name of the first author, we have identified one duplicate. In 2006, Burdof published two articles using the same multi-state model4,5: however, the dataset they used for both publications are different. The first article uses data from a meta-analysis (and was published in a journal specialized in vibration) and the second one data from a cohort (and was published in a generalist occupational health journal). 

• The determinants found to be likely to influence SL might be correlated to one another, such as physical issues and job types.

We thank the reviewer for this remark: this is one of the most important issues with sick leave data and we should have assessed this point more precisely. 

Even if it is a key issue, this point is seldom discussed in the literature since most of the times, few covariates are used and the correlations do not really impact the methodology in this case, it will mostly impact the interpretation of the results. We had nevertheless noted an article using variable selection6 : (however we did not report systematically this point so there may have been other articles that use these methods). This article used a backward selection to choose the most relevant variable. This allows for a more readable model but do not really solve the problem of correlations. If variables are highly correlated, the model will simply choose the most significant variables from this bunch of correlated variables. Variable selection is necessary if there are too many covariates but must always be coupled with a good correlation analysis.

 The discussion of correlation is indeed generally done in the interpretation of the results. In particular, we see in Duchemin et al.7 that there is a strong correlation between age and perceived health: age very often emerges as a determining factor in long-term cessation but, once health is controlled for, it no longer emerges. 

We added this discussion in the manuscript in the last paragraph of our discussion that dealt lightly with this subject (lines 250-258) and thank again the reviewer for this remark.

References

1. Brown J, Mackay D, Demou E, Craig J, Frank J, Macdonald EB. The EASY (Early Access to Support for You) sickness absence service: a four-year evaluation of the impact on absenteeism. Scand J Work Environ Health. 2015;41(2):204-215. doi:10.5271/sjweh.3480

2. Spears DR, McNeil C, Warnock E, et al. Predicting Temporal Trends in Sickness Absence Rates for Civil Service Employees of a Federal Public Health Agency. J Occup Environ Med. 2013;55(2):179-190. doi:10.1097/JOM.0b013e3182717eb5

3. Markussen S, Røed K. Daylight and absenteeism – Evidence from Norway. Economics & Human Biology. 2015;16:73-80. doi:10.1016/j.ehb.2014.01.002

4. Burdorf A, Hulshof CTJ. Modelling the effects of exposure to whole-body vibration on low-back pain and its long-term consequences for sickness absence and associated work disability. Journal of Sound and Vibration. 2006;298(3):480-491. doi:10.1016/j.jsv.2006.06.023

5. Burdorf A, Jansen JP. Predicting the long term course of low back pain and its consequences for sickness absence and associated work disability. Occup Environ Med. 2006;63(8):522-529. doi:10.1136/oem.2005.019745

6. Alavinia SM, van den Berg TI, van Duivenbooden C, Elders LA, Burdorf A. Impact of work-related factors, lifestyle, and work ability on sickness absence among Dutch construction workers. Scandinavian Journal of Work, Environment & Health. 2009;35(5):325-333. doi:10.5271/sjweh.1340

7. Duchemin T, Bar-Hen A, Lounissi R, Dab W, Hocine MN. Hierarchizing Determinants of Sick Leave: Insights From a Survey on Health and Well-being at the Workplace. Journal of Occupational and Environmental Medicine. 2019;61:1. doi:10.1097/JOM.0000000000001643

---

## [Editor Report · Decision Letter 1]

28 Aug 2020

Modeling sickness absence data: a scoping review

PONE-D-19-24227R1

Dear Dr. Duchemin,

We’re pleased to inform you that your manuscript has been judged scientifically suitable for publication and will be formally accepted for publication once it meets all outstanding technical requirements.

Kind regards,

Bhaskaran Unnikrishnan, MD

Academic Editor

PLOS ONE

Additional Editor Comments (optional):

Accepted
---

## [Editor Report · Acceptance letter]

2 Sep 2020

PONE-D-19-24227R1 

Modeling sickness absence data: a scoping review 

Dear Dr. Duchemin:

I'm pleased to inform you that your manuscript has been deemed suitable for publication in PLOS ONE. Congratulations! Your manuscript is now with our production department. 

Kind regards, 

on behalf of

Dr. Bhaskaran Unnikrishnan 

Academic Editor

PLOS ONE